# Dysautonomia, but Not Cardiac Dysfunction, Is Common in a Cohort of Individuals with Long COVID

**DOI:** 10.3390/jpm13111606

**Published:** 2023-11-14

**Authors:** Laura Tabacof, Jamie Wood, Erica Breyman, Jenna Tosto-Mancuso, Amanda Kelly, Kaitlyn Wilkey, Chi Zhang, David Putrino, Amy Kontorovich

**Affiliations:** 1Abilities Research Center, Department of Rehabilitation and Human Performance, Icahn School of Medicine at Mount Sinai, New York, NY 10029, USAjamie.wood@mountsinai.org (J.W.);; 2Department of Rehabilitation and Human Performance, Icahn School of Medicine at Mount Sinai, New York, NY 10029, USA; 3Zena and Michael A. Weiner Cardiovascular Institute, Icahn School of Medicine at Mount Sinai, New York, NY 10029, USA; chi.zhang2@mountsinai.org (C.Z.); amy.kontorovich@mountsinai.org (A.K.); 4The Mindich Child Health and Development Institute, Icahn School of Medicine at Mount Sinai, New York, NY 10029, USA; 5The Cardiovascular Research Institute, Icahn School of Medicine at Mount Sinai, New York, NY 10029, USA

**Keywords:** postural orthostatic tachycardia syndrome (POTS), dysautonomia, long COVID, echocardiogram, post-acute sequelae of COVID, PASC

## Abstract

Despite the prevalence of dysautonomia in people with Long COVID, it is currently unknown whether Long COVID dysautonomia is routinely accompanied by structural or functional cardiac alterations. In this retrospective observational study, the presence of echocardiographic abnormalities was assessed. Left ventricular (LV) chamber sizes were correlated to diagnostic categories and symptoms via standardized patient-reported outcome (PRO) questionnaires. A total of 203 individuals with Long COVID without pre-existing cardiac disease and with available echocardiograms were included (mean age, 45 years; 67% female). Overall, symptoms and PRO scores for fatigue, breathlessness, quality of life, disability, anxiety and depression were not different between those classified with post-COVID dysautonomia (PCD, 22%) and those unclassified (78%). An LV internal diameter at an end-diastole z score < −2 was observed in 33 (16.5%) individuals, and stroke volume (SV) was lower in the PCD vs. unclassified subgroup (51.6 vs. 59.2 mL, 95% C.I. 47.1–56.1 vs. 56.2–62.3). LV end-diastolic volume (mean diff. (95% CI) −13 [−1–−26] mL, *p* = 0.04) and SV (−10 [−1–−20] mL, *p* = 0.03) were smaller in those individuals reporting a reduction in physical activity post-COVID-19 infection, and smaller LVMI was weakly correlated with worse fatigue (r = 0.23, *p* = 0.02). The majority of individuals with Long COVID report shared symptoms and did not demonstrate cardiac dysfunction on echocardiography.

## 1. Introduction

Although SARS-CoV-2 infection has been phenomenologically linked with both cardiac abnormalities [1,2,3,4,5,6,7,8] and persistent lingering symptoms referred to as Long COVID [9,10], it remains unknown whether these two sets of complications are pathologically connected in a majority of cases. Cardiac effects from COVID-19 infection have been described within several categories: (1) myocardial injury seen in ~30% of acutely hospitalized individuals, as evidenced by elevated cardiac biomarkers (i.e., troponin) [1]; (2) coagulopathy leading to microvascular or macrovascular thromboembolic events [2,3]; (3) rare cases of clinical myocarditis or myopericarditis necessitating hospitalization either in adults or in children in the context of Multisystem Inflammatory Syndrome in Children (MIS-C) [4,5,6]; and (4) cardiac MRI evidence of inflammation, identified primarily in prospective surveys of those with mild initial COVID-19 illness, such as young athletes, which is typically limited and transient [7,8]. In distinct contrast to these COVID-related cardiac endotypes, Long COVID is estimated to affect as many as 20% of individuals who have survived an initial acute COVID-19 infection [11] and leads to diverse chronic symptoms, typically without objective abnormalities on standard-of-care diagnostic laboratory or imaging tests [9]. Symptoms include those that may be considered referable to the cardiovascular system (chest discomfort, palpitations, breathlessness and exercise intolerance) as well as systemic (fatigue, brain fog, dizziness, memory loss and weakness) [9,10]. Many individuals experience debilitating symptoms for more than 12 months following acute infection [9]. 

A cornerstone of Long COVID is post-exertional symptom exacerbation following an increase in levels of physical or mental exertion [9,10]. This feature is also cardinal to a set of conditions that fall within the classification of autonomic nervous system dysfunction or dysautonomia. Indeed, dysautonomia has been described in Long COVID [12,13,14,15], and the reported causes of symptom exacerbation (physical exertion, stress, dehydration, weather changes, consuming large meals, premenstrual period and alcohol) and reductions in levels of physical activity are shared between Long COVID and other forms of dysautonomia [9,10,12,16,17]. Most individual presentations of dysautonomia in Long COVID span multiple subtypes of autonomic dysfunction, with up to 33% meeting criteria for specific diagnoses such as postural orthostatic tachycardia syndrome (POTS) [18,19,20]. Dysautonomia is not classically associated with myocardial inflammation or dysfunction. However, given the prevalence of myocardial involvement as a sequela of COVID-19, it was sought to determine whether post-acute COVID dysautonomia (PCD) is associated with distinct cardiac abnormalities, including features of cardiac atrophy, as reported in other forms of dysautonomia [16,17]. In this study, symptom burden and echocardiographic findings among a “real world,” richly phenotyped Long COVID/post-acute COVID-19 dysautonomia (PCD) cohort were analyzed retrospectively.

This report describes the findings from routine echocardiogram assessments obtained from a cohort of individuals attending a Long COVID clinic, and explores the presence of clinically diagnosed PCD alongside self-reported persistent symptoms. 

## 2. Materials and Methods

### 2.1. Study Design

This was an observational study using retrospectively obtained electronic health record (EHR) information and patient-reported outcomes (PROs). Approval for publication was provided by the Mount Sinai Program for Protection of Human Subjects (IRB 21-00944). A waiver of consent was approved.

### 2.2. Participants

Adults attending the Long COVID clinic at Mount Sinai Hospital were included. The Long COVID clinic is an interdisciplinary clinic consisting of physicians (primary care and a range of subspecialties including physiatry and cardiology), physical therapists, dietitians and researchers. 

Inclusion criteria were EHR diagnosis of Long COVID, defined as experiencing new, returning or ongoing health problems 4 or more weeks following initial COVID-19 infection in the absence of any specific organ damage using standard clinical testing protocols [9,21], and having a transthoracic echocardiogram (TTE) assessment performed at Mount Sinai Hospital > 28 days following diagnosis with COVID-19. Individuals were excluded if they had a diagnosis of heart failure, cardiomyopathy or dysautonomia prior to COVID-19 infection.

### 2.3. Data Collection and Outcomes

Data including demographics, acute COVID-19 hospitalization status, need for mechanical ventilation and duration of COVID-19-related symptoms were obtained retrospectively from the EHRs as well as a patient-reported outcome (PRO) surveys developed by Long COVID clinic team members and administered as part of clinical care.

### 2.4. Echocardiographic Assessment

Echocardiographic data were systematically extracted from clinical reports through a Mount Sinai Data Warehouse query and included left ventricular (LV) and right ventricular (RV) size and function; LV ejection fraction (EF); left and right atrial sizes; qualitative descriptors of the aortic, mitral, tricuspid and pulmonic valve anatomy and function; presence/absence of pulmonary hypertension; thoracic aortic dilatation; pericardial abnormalities; LV internal diameter at end diastole (LVIDd) and end systole (LVIDs); LV end-diastolic volume (LVEDV), end-systolic volume (LVESV) and stroke volume (SV) from the four-chamber apical view; and LV mass index (LVMI). The LVIDd and LVIDs z scores were calculated using the formula: (LVID_measured_ − LVID_mean_)/SD, with normal mean and SD values obtained from the American Society of Echocardiography and the European Association of Cardiovascular Imaging [22]. Additionally, the following quantification methods were used for other outcomes: Teicholz 2D (LVIDd) and biplane Simpson (LVEDV 4C, LVESV 4C, EF); SV 4C was calculated using 0.785 × left ventricular outflow tract (LVOT) diameter2 × LVOT velocity time integral; LV mass index was calculated using 0.8 (1.04 ([LVIDd + posterior wall thickness in diastole + interventricular septum thickness in diastole]3 − [LVIDd]3))+ 0.6 g; and LV mass index was calculated using LV mass/BSA.

Abnormalities of valves were defined as either structural abnormalities or moderate or higher valvular stenosis or insufficiency. Echocardiographic reports were manually over-read by the investigators (C.Z., A.K.) to ensure completeness of the data. Echocardiograms performed prior to COVID-19 infection were also reviewed if available.

### 2.5. Classification of Clinically Diagnosed Dysautonomia

EHRs (primary care or cardiology progress notes, “problem list”) were manually reviewed to identify whether individuals with Long COVID were diagnosed with dysautonomia/PCD specifically as documented by their treating cardiologist or physician or were otherwise “unclassified” (no stated diagnosis of dysautonomia in the EHR). Participants were classified by the research team as having dysautonomia if their treating cardiologist or physician documented this diagnosis in the EHR, with supporting evidence including symptoms and clinical/historical features, and/or formal testing including a tilt table test, active stand test, quantitative sudomotor axon reflex test, thermoregulatory sweat test, or using heart rate variability. Members of the research team were not involved in the initial diagnosis of dysautonomia.

### 2.6. Patient-Reported Outcomes

Individuals attending the Long COVID clinic were requested to complete a PRO survey as part of their routine clinical care. Survey data were collected using Research Electronic Data Capture (REDCap) electronic data capture tools hosted in the Mount Sinai Health System. Participants were provided with a survey link via email to complete online. The PROs included persistent symptoms and triggers of symptom exacerbation and screening tools for breathlessness (Medical Research Council (MRC) Breathlessness Scale), health-related quality of life (HRQoL) (EuroQol EQ-5D-5L), fatigue (Fatigue Severity Scale (FSS), fatigue visual analog scale (VAS)), completion of regular-, moderate- and vigorous-intensity physical activity (author developed), cognitive function (Neuro-QOL), anxiety (GAD-7), depression (PHQ-2) and disability (WHODAS).

### 2.7. Statistical Plan

Statistical analyses were undertaken with Stata (StataCorp, Stata Statistical Software Release: V.14). Data are presented as frequencies and proportions, mean and standard deviation (SD) or median and 95% confidence interval (CI) where appropriate. Correlations between echocardiographic variables and PROs were examined using Pearson’s correlation or Spearman’s correlation where appropriate. Independent sample t-tests were used to examine between-group differences based on the classification of dysautonomia, presented as mean difference and 95% CI. Proportions were examined using Pearson’s chi-squared test, presented as frequency (%).

## 3. Results

Participants were identified from a database of 737 individuals with Long COVID. Of these, 217 (29%) had an echocardiogram performed in a clinical setting at least 28 days following their COVID-19 infection (Figure 1). Fourteen (6%) were excluded from analyses as they had pre-existing (i.e., prior to COVID-19 infection) diagnosed heart failure, cardiomyopathy and/or dysautonomia. The majority of the final cohort were female (67%) and did not require hospitalization during acute COVID-19 infection (Table 1). 

Echocardiographic abnormalities among the 203 included individuals are presented in Table 2. LV and RV systolic function was normal in 201 (99%) and 203 (100%) individuals, respectively. Of the 202 (99%) individuals for whom an EF was reported, the median (95% confidence interval (CI)) value was 65 ± 5%. Diastolic dysfunction was present in 8 (4%), LV hypertrophy in 20 patients (11%), pulmonary hypertension in 2 (1%; 1 also with RV dilatation) and pericardial abnormalities in 3 (1%) individuals. These abnormalities were new (n = 2, 7, 1, 2), documented prior to COVID-19 infection (n = 0, 2, 0, 1) or prior echocardiograms were unavailable (n = 6, 11, 1, 0) in affected individuals. Three (1%) individuals were diagnosed with myocardial impairments attributed to COVID-19 infection: one patient with segmental wall motion abnormality with borderline LV EF (no prior echocardiogram available for comparison), one patient with normal LV size and diffuse borderline LV systolic dysfunction (EF 50%; no prior echocardiogram available) and one with severe LV dilatation and EF 56% (new compared to pre-COVID-19 echocardiogram). Since Long COVID was defined as cases without structural cardiac changes that could otherwise explain persistent symptoms, these individuals were excluded from downstream analyses. 

Of the 200 individuals with Long COVID included in the secondary analyses, 44 (22%) were classified by their provider (primary care or cardiologist) as having PCD; classifications were made based on symptoms and clinical/historical features (n = 28 (64%)) and/or formal testing (n = 17 (39%); these included tilt table test, active stand test, heart rate variability, quantitative sudomotor axon reflex test and/or thermoregulatory sweat test). The remaining 156 (78%) were labeled as having Long COVID without further classification. 

Stroke volumes (SVs) were lower in the PCD group when compared to those not classified (Table 3). A similar subset of individuals in both the PCD (n = 10, 23%) and unclassified groups (n = 23, 15%) had LVIDd measurements smaller than sex-specific normal expected values (Z-score ≤ −2, *p* = 0.25) [22]. Overall, there were no differences in echocardiographic LV chamber size measures between the two groups.

Of the 200 individuals with echocardiographic measures reported, 99 (50%) completed the PRO survey. The mean (95% CI) total number of symptoms reported was 12 (1 to 33) for individuals with PCD, as well as individuals who were unclassified. Scores from PROs screening for fatigue, breathlessness, quality of life, disability, anxiety and depression were not different between those with PCD and those who were unclassified (Table 4).

The majority of individuals who completed the physical activity survey questions reported a reduced engagement in moderate-intensity (69/85 (81%)) and vigorous-intensity (60/86 (70%)) physical activity post-acute COVID-19 infection. Both LVEDV (mean diff. (95% CI) −13 [−1–−26] mL, *p* = 0.04) and SV (−10 [−1–−20] mL, *p* = 0.03) measurements were smaller in those reporting reduced engagement in moderate-intensity physical activity post-COVID-19 infection when compared to those with similar or more regular engagement in moderate-intensity physical activity. Smaller LVMI measurements were weakly correlated with worse fatigue VAS scores (r = 0.23, *p* = 0.02). 

## 4. Discussion

Both myocardial injury and the phenotype of Long COVID are important known consequences of infection with SARS-CoV-2. However, it is not yet established whether these phenomena are epidemiologically or pathobiologically related. Because Long COVID symptoms classically include some combination of fatigue, palpitations, breathlessness, chest discomfort and post-exertional symptom exacerbation, it is imperative to clarify whether these are manifestations of cardiac dysfunction. These data demonstrate that in the overwhelming majority of a set of well-characterized individuals evaluated in a Long COVID clinic, LV and RV function is normal. Of this cohort, only three individuals had evidence of new-onset cardiomyopathy (segmental or diffuse LV dysfunction and/or LV dilatation). Outside of these edge cases, the majority of individuals with Long COVID in this study did not require hospitalization for acute COVID-19, and it was found that their persistent symptoms were not associated with structural or functional cardiac abnormalities. These findings support that the phenomena of SARS-CoV-2-induced myocardial injury and Long COVID are not directly related. As such, a cardiac rehabilitation approach that might be appropriate following myocardial infarction or heart failure hospitalization is unlikely to be effective in treating Long COVID symptoms and should not be prescribed.

Previous data have shown that the typical Long COVID presentation incorporates features of dysautonomia, with individuals reporting systemic symptoms (fatigue, breathlessness, chest discomfort, palpitations, dizziness, syncope/presyncope, orthostatic exacerbations, leg pain and exercise intolerance/post-exertional symptom exacerbation). Further, the triggers of worsening symptoms (physical exertion, stress, dehydration, weather changes, consuming large meals, premenstrual period and alcohol) are similar [9]. Dysautonomia is now an established endotype of Long COVID [12,13,14]; yet, only 13–33% of individuals meet the criteria for specific diagnoses such as POTS [18,19,20]. It is, therefore, necessary to define a new terminology: post-COVID-19 dysautonomia (PCD).

The proposed pathophysiologic connections between COVID-19 and dysautonomia include (1) hypovolemia due to fever, decreased fluid intake, nausea, excessive diaphoresis and prolonged bed rest, leading to increased cardiac sympathetic nervous system outflow and cardiac atrophy; (2) direct SARS-CoV-2 infection and the destruction of extracardiac postganglionic sympathetic neurons, increasing sympathetic outflow; (3) SARS-CoV-2 invasion of the brainstem, resulting in increased central sympathetic outflow analogous to that seen in Takotsubo cardiomyopathy; and (4) virally induced autoimmunity, directing an immune attack against host neurons [23]. Further work is required to better understand which, if any, of these mechanisms underlie PCD.

Small cardiac size with reduced blood volume (i.e., cardiac atrophy) has previously been noted in POTS [17] and other forms of dysautonomia [16]. Here, it is reported that ~17% of individuals with Long COVID demonstrate features of small LV chamber size and that echocardiographic signs of cardiac atrophy correlate with reductions in moderate physical activity (lower LV EDV and SV) and worse fatigue (smaller LVMI). Clinicians should be attuned to recognizing PCD, as affected individuals may benefit from a number of effective interventions (i.e., oral fluid and electrolyte repletion as well as lower extremity compression garments), pharmacotherapy and/or autonomic rehabilitation [24]. The latter is a specialized form of rehabilitation that focuses on utilizing symptom-titrated exercises that focus on retraining appropriate physiological responses to autonomic challenges, and can improve or resolve symptoms [25,26] while increasing cardiac mass and blood volumes in a majority of cases [17,27]. Among the Long COVID cohort, only 22% of individuals were diagnosed with PCD, even though symptom type, severity and LV chamber size were not distinguishable from the unclassified group. Overall, significant overlap in clinical and cardiac profiles between the PCD and unclassified groups indicates that these represent a single phenotype, with the etiology of symptoms being appropriately recognized in only a fraction of LC patients. Only SV was lower in the PCD group, suggesting that there may have been features present in these individuals that raised suspicion for dysautonomia or prompted a work-up with formal autonomic testing. For example, individuals with lower SV may show physiologic signs such as sinus tachycardia or exaggerated exertional or orthostatic tachycardia.

For many years, dysautonomia (that has been triggered by conditions or events other than acute SARS-CoV-2 infection) has been associated with reduced cardiac size, and improvements in functional testing after rehabilitation have been associated with complementary increases in cardiac size [16,17]. This has led to the narrative that at least some dimension of the disability caused by dysautonomia may be caused by “cardiovascular deconditioning” [27]. However, few studies account for the fact that prior to the emergence of Long COVID, dysautonomia was not diagnosed until, on average, almost six years after the initial emergence of symptoms [28]. At this time, it is possible that significant deconditioning can occur as otherwise healthy and active individuals begin to avoid physical activity due to the propensity to produce post-exertional symptoms. With the emergence of Long COVID as a chronic post-acute infection syndrome and the observation that a large proportion of Long COVID cases are accompanied by dysautonomia [29], cases of PCD are being diagnosed much faster than dysautonomia that was diagnosed prior to 2020. In this study, there was a shorter period of time between the onset of PCD symptoms and its diagnosis (since all diagnoses of PCD in this study happened in under 2 years of symptom onset). This study’s finding of minimal changes in cardiac size related to a diagnosis of PCD compared with those without a PCD diagnosis would indicate that such cardiac changes are in fact related to prolonged periods of inactivity, rather than any cardinal pathobiological features of dysautonomia. Few studies in the field have had the opportunity to study cardiac morphology of cases of dysautonomia that have been diagnosed so soon after the onset of symptoms.

The limitations of this study include the use of data collected retrospectively from a convenience sample and the lack of baseline information, such as pre-COVID-19 PROs and echocardiographic features for most individuals. The classification of dysautonomia by the research team relied on EHR information, which may result in issues such as variability between cardiologists and physicians in their assessment and diagnosis of dysautonomia, a lack of uniformity in the outcomes used to confirm a diagnosis dysautonomia including electrocardiograms and the potential for information to be absent from the EHR for both classified and unclassified groups.

The contemporary absence of universal diagnostic criteria for dysautonomia and, more specifically, PCD challenges the ability to identify individuals who would benefit from interventions. The improved recognition of PCD can address important disparities in healthcare, especially for women, who are more frequently affected. Although cohort-based observations support cardiac atrophy as a mechanism contributing to PCD, echocardiographic measurements are not likely to be useful as sole biomarkers to capture this phenotype. Further studies are needed to more rigorously define PCD and enable improved recognition and care.

## 5. Conclusions

The majority of individuals with Long COVID report shared symptoms and did not demonstrate cardiac dysfunction on echocardiography. Cardiac atrophy, as has been previously reported in association with other forms of dysautonomia, is a feature of Long COVID and correlates with reductions in physical activity levels and worse fatigue. Improved biomarkers of PCD are needed to enable better recognition and care for patients with Long COVID.

## Figures and Tables

**Figure 1 jpm-13-01606-f001:**
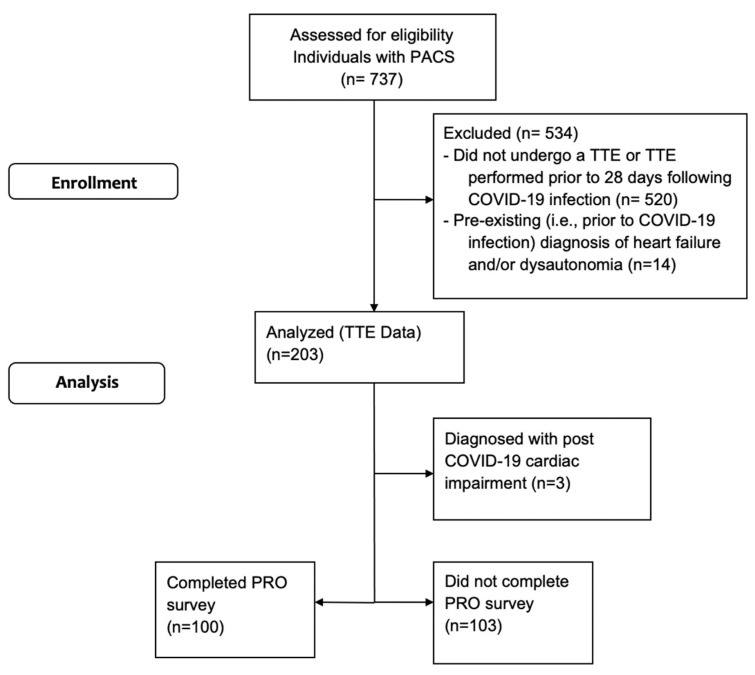
Consort flow diagram.

**Table 1 jpm-13-01606-t001:** Demographics of individuals with Long COVID (n = 203).

	All Participants (n = 203)	Female (n = 135)	Male (n = 68)	Dysautonomia (n = 45)	Unclassified (n = 158)
Female, n (%)	135 (67%)	135 (100)	0 (0)	37 (82)	98 (62)
Age, y	45 (22–80)	46 (22–79)	44 (23–80)	47 (23–79)	45 (22–80)
BSA	1.9 (1.3–2.6)	1.8 (1.3–2.5)	2.1 (1.6–2.6)	1.8 (1.4–2.3)	1.9 (1.3–2.6)
Hospitalized for COVID-19, n (%)	44 (22)	29 (21)	15 (22)	8 (18)	36 (23)
Mechanical ventilation, n (%)	5 (2)	5 (4)	0 (0)	0 (0)	8 (18)
Duration of symptoms, days	218 (34–500)	221 (36–479)	213 (34–500)	212 (36–479)	220 (34–500)

Data are presented as mean (range) otherwise indicated. BSA—body surface area. PCR—polymerase chain reaction. Mechanical ventilation—if required, mechanical ventilation was utilized during acute COVID-19 infection.

**Table 2 jpm-13-01606-t002:** Echocardiographic abnormalities in individuals with Long COVID (n = 203).

	Parameter Reported (%)	Abnormal (%)
LV systolic function	203 (100)	2 (1)
LV diastolic function	189 (93)	8 (4)
LV hypertrophy	179 (88)	20 (11)
LV dilatation	203 (100)	1 (0.5)
RV systolic function	197 (97)	0 (0)
RV size	197 (97)	2 (1)
Pulmonary hypertension	77 (38)	2 (3)
LA size	127 (63)	4 (3)
RA size	127 (63)	1 (1)
Mitral valve	200 (99)	1 (1)
Aortic valve	200 (99)	1 (1)
Tricuspid valve	200 (99)	0 (0)
Pulmonic valve	200 (99)	0 (0)
Pericardium	113 (56)	3 (3)
Aortic root dilatation	85 (42)	3 (4)
Miscellaneous	-	3 (1)

Data are reported as number and percentage. LV—left ventricle; RV—right ventricle; LA—left atrium; RA—right atrium. Mitral valve abnormality was mitral valve prolapse with mild-to-moderate insufficiency. Aortic valve abnormality was bicuspid aortic valve. Miscellaneous abnormalities included interatrial septal aneurysm (n = 1); lipomatous atrial septal hypertrophy (n = 1); ventricular septal defect versus sinus of Valsalva aneurysm (n = 1).

**Table 3 jpm-13-01606-t003:** Echocardiographic measures in individuals with Long COVID (n = 200).

	All Participants (n = 200)	Dysautonomia (n = 44)	Unclassified (n = 156)	Difference
LVIDd z score	−0.89 (−1.06–−0.72)	−1.16 (−1.55–−0.77)	−0.81 (−1.00–−0.62)	0.35 (−0.06–0.72)
LVIDs z score ^a^	−0.42 (−0.57–−0.28)	−0.50 (−0.82–−0.18)	−0.40 (−0.57–−0.18)	0.10 (−0.26–0.45)
LVEDV ^a^	89.0 (85.3–92.5)	82.5 (75.5–89.4)	90.8 (86.6–94.9)	8.3 (−0.3–16.9)
LVESV ^a^	39.3 (33.8–44.7)	30.0 (25.0–35.0)	42.2 (35.2–49.1)	12.2 (−0.5–24.9)
SV ^a^	57.5 (54.9–60.1)	51.6 (47.1–56.1)	59.2 (56.2–62.3)	7.6 (1.6–13.7)
LVMI	85.2 (82.4–88.0)	85.8 (79.3–92.3)	85.0 (82.0–88.1)	0.8 (−7.5–5.9)

Data are presented as mean (95% confidence interval). ^a^ Echocardiographic measurements not available for all participants. LVID z score: dysautonomia, n = 43; not classified, n = 150. LVEDV: dysautonomia, n = 37; not classified, n = 130. LVESV: dysautonomia, n = 23; not classified, n = 73. SV: dysautonomia, n = 35; not classified, n = 116.

**Table 4 jpm-13-01606-t004:** Patient-reported outcomes for individuals (n = 99) who completed the survey.

Patient-Reported Outcome	All Participants (n = 99)	Dysautonomia (n = 27)	Unclassified (n = 72)	Difference (95% CI)
MRC breathlessness scale	2 (1–4)	2 (1–4)	2 (1–4)	0 (−1–0)
EQ-5D-5L domains				
Mobility	2 (1–4)	2 (1–4)	2 (1–4)	0 (−1–0)
Usual activities	3 (1–5)	3 (1–5)	3 (1–5)	0 (−1–1)
Anxiety/depression	3 (1–4)	3 (1–4)	3 (1–4)	0 (−1–1)
Self-care	1 (1–3)	1 (1–3)	1 (1–3)	0 (−1–0)
Pain/discomfort	3 (1–4)	3 (1–4)	2 (1–4)	−1 (−1–0)
EQ-5D-5L health status VAS ^a,b^	59 (55–63)	52 (43–60)	61 (57–66)	10 (1–19)
Fatigue Severity Scale, total score ^a,b^	49 (46–51)	52 (48–56)	48 (44–51)	−4 (−10–2)
Fatigue VAS (0 to 100) ^a,b^	44 (39–50)	43 (32–55)	45 (38–51)	2 (−11–14)
Neuro-QOL, t score ^a,b^	41 (39–44)	39 (34–44)	42 (39–45)	3 (−3–9)
GAD-7, total score ^a^	7 (6–8)	8 (5–10)	6 (5–8)	−1 (−4–1)
PHQ-2, total score ^a^	2 (2–2)	2 (2–3)	2 (2–2)	0 (−1–0)
WHODAS, total score ^a,b^	32 (28–37)	37 (29–45)	30 (25–36)	−7 (−17–4)

Data are presented as median (95% confidence interval (CI)) or ^a^ mean (95% CI). ^b^ EQ-5D-5L health status VAS, fatigue VAS: dysautonomia, n = 26; not classified, n = 70. Fatigue Severity Scale, GAD-7: dysautonomia, n = 26; not classified, n = 71; Neuro-QOL: dysautonomia, n = 16; not classified, n = 54; WHODAS: dysautonomia, n = 24; not classified, n = 65.

## Data Availability

Data supporting reported results can be provided upon reasonable request to the corresponding author.

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
