# Peer review of "Dysautonomia, but Not Cardiac Dysfunction, Is Common in a Cohort of Individuals with Long COVID"

_jpm, 2023, doi:10.3390/jpm13111606_

Round 1
Reviewer 1 Report
Comments and Suggestions for Authors
The idea of the manuscript is very fresh, actual and important for the clinical course of long COVID-19 infection, but there are many mistakes in metodology, unfinished results and noncomparabiled variables. Those results did not showed whether long COVID-19 symptoms are manifestations of cardiac dysfunction.
In methodology There is no ECG data including HR, which is one of the major distinguishing signs for PCD. Measurement approach for LV dimensions and functions including global EF are not clearly described (Simpson rule, Teicholz?). Available previous echo data should be necessery for comparision in all patinets. The is no clear explanation about Z score? Tissue tracking GLS values could be more sensitive for asssessing global systolic LV function and possibly more predictive in patients with long COVID-19.
Comments on the Quality of English Language
quality of English language is solid
Author Response
The idea of the manuscript is very fresh, actual and important for the clinical course of long COVID-19 infection, but there are many mistakes in methodology, unfinished results and noncomparable variables. Those results did not show whether long COVID-19 symptoms are manifestations of cardiac dysfunction.
In methodology: There is no ECG data including HR, which is one of the major distinguishing signs for PCD.
It was not feasible to access ECG data for all participants, and this limitation has now been noted in the manuscript.
Measurement approach for LV dimensions and functions including global EF are not clearly described (Simpson rule, Teicholz?).
The measurement approach has been expanded in the methods.
Available previous echo data should be necessary for comparison in all patients. The is no clear explanation about Z score?
Access to previous echo data was not available for most of the participants and therefore was unable to be included; this is already listed as a limitation. The Z score calculation is described in the methods.
Tissue tracking GLS values could be more sensitive for assessing global systolic LV function and possibly more predictive in patients with long COVID-19.
The authors acknowledge that using tissue tracking (speckle tracking echocardiography) GLS values may add more useful information to the manuscript. Speckle tracking echocardiography is not typically performed as part of a standard routine clinical echocardiogram and therefore the majority of the echocardiograms included for this study did not include strain assessments. We therefore could not uniformly analyze strain for this study. While some echocardiography software packages permit retrospective strain analysis, clinical echocardiograms included in this study were performed at multiple hospital and clinical sites using several different ultrasound makes and models, thus it was not possible to retrospectively calculate strain and meaningfully compare results across studies.
Reviewer 2 Report
Comments and Suggestions for Authors
Dear Authors,
As a cardiologist, I was interested in reviewing your paper. Our clinical experience provided many examples of Long COVID. Despite clinical observations, Long COVID is a new condition which is still being studied.
From this point of view I consider that your paper is a very useful signal for clinicians. Moreover, this study has some strengths as follows:
- it has a clear and innovative objective namely the analysis of post-acute COVID dysautonomia (PCD) association with structural or functional cardiac alterations;
- the methodology is clearly and well-described. There are defined the echocardiographic parameters which are related to "reductions in moderate physical activity (lower LV EDV and SV) and worse fatigue (smaller LVMI)". Also, cardiac atrophy is demonstrated to be a feature of Long COVID.
- the results are synthetized in suggestive tables, the discussion is balanced and emphasizes the limitations of the study;
- the authors suggest the pathophysiologic interplay between COVID-19 and dysautonomia;
- the study has an obvious practical value as it is highlighted the importance of recognizing PCD, which benefits from effective therapeutic methods as well as from a specialized form of rehabilitation;
- the relative small number of the selected references demonstrates the scarce information in the field.
So, I think this paper is an excellent synthesis of the problem studied.
Author Response
Thank you for this positive review.
Reviewer 3 Report
Comments and Suggestions for Authors
Dear authors,
In this paper entitled " Dysautonomia, but not cardiac dysfunction, is common in a cohort of individuals with Long CCOVID" you try to describe indirectly the dysautonomia in patients with long covid syndrome. Particularly, you focused on the POTS diagnosis based on the ecocardiographic parameters.
In my opinion, in this paper the methods lacks on the the tests mandatory for the diagnosis of the dysautonomic disorders, like: tilt test, cold face test, hand grip test, QSART and many others.
In the other hands, is uncorrect to discuss of the dysaunomia without any test that define not only the presence or absence of the authonomic system involvement,.but also the subtype of dysautonomic dysfunction.
I suggest or to change te title of the paper, adding "indirect dysautonomia diagnosis etc" or to re-write the study design (for exemple admnistering some.specif questionnaire on dysautonomic synthoms to the patients). The last option is the better in my opinion.
Author Response
In this paper entitled " Dysautonomia, but not cardiac dysfunction, is common in a cohort of individuals with Long CCOVID" you try to describe indirectly the dysautonomia in patients with long covid syndrome. Particularly, you focused on the POTS diagnosis based on the echocardiographic parameters.
In my opinion, in this paper the methods lacks on the the tests mandatory for the diagnosis of the dysautonomic disorders, like: tilt test, cold face test, hand grip test, QSART and many others. In the other hands, is uncorrect to discuss of the dysaunomia without any test that define not only the presence or absence of the autonomic system involvement, but also the subtype of dysautonomic dysfunction.
Please see our earlier response to the Editor, and the additions made to the methods and limitations regarding the diagnosis and classification of dysautonomia in this research. The research team were not making a diagnosis of dysautonomia, and were relying on the assessment of trained cardiologists and physicians familiar with American autonomic society evaluation criteria for making a diagnosis of dysautonomia. Any objective testing used by the diagnosing clinicians were listed in the methods and results.
I suggest or to change te title of the paper, adding "indirect dysautonomia diagnosis etc" or to re-write the study design (for exemple admnistering some specific questionnaire on dysautonomic synthoms to the patients). The last option is the better in my opinion.
With the clarifications of diagnosis that we have provided, we hope to keep the title of the manuscript as is and will defer to the Editor if this needs to be changed.
Round 2
Reviewer 1 Report
Comments and Suggestions for Authors
Methodology: better explanation of the term "cardaic atrophy", because it is unkonown to echocardiographers
In Reesults: Table 1 and legend are not entirely clear.
Another limitation, echocardiograms of the patients before inclusion in the study were not available.
Comments on the Quality of English Language
English could be more clearer without long senteces where the point is lost
Reviewer 3 Report
Comments and Suggestions for Authors
I accept in this modified form
